# Drought Stress Tolerance and Photosynthetic Activity of Alloplasmic Lines *T. dicoccum* x *T. aestivum*

**DOI:** 10.3390/ijms21093356

**Published:** 2020-05-09

**Authors:** Nina V. Terletskaya, Andrey B. Shcherban, Michail A. Nesterov, Roman N. Perfil’ev, Elena A. Salina, Nazira A. Altayeva, Irina V. Blavachinskaya

**Affiliations:** 1Department of Biodiversity and Biological Resources, Faculty of Biology and Biotechnology Al-Farabi Kazakh National University, Al-Farabi av., 71, Almaty 050040, Kazakhstan; irina-b-1952@mail.ru; 2Institute of Plant Biology and Biotechnology, Timiryazev str. 45, Almaty 050040, Kazakhstan; daizy-c@mail.ru; 3Kurchatov Genomics Center, Institute of Cytology and Genetics SB RAS, Lavrentiev av., 10, 630090 Novosibirsk, Russia; atos@bionet.nsc.ru (A.B.S.); mikkanestor@bionet.nsc.ru (M.A.N.); salina@bionet.nsc.ru (R.N.P.); sunday01@mail.ru (E.A.S.); 4Central Laboratory for Biocontrol, Certification and Preclinical Trials, Al-Farabi av., 93, Almaty 050040, Kazakhstan

**Keywords:** alloplasmic wheat lines, drought tolerance, photosynthesis, *DREB*, *orf256*, *rps19-p* genes, SSR

## Abstract

Tetraploid species *T. dicoccum* Shuebl is a potential source of drought tolerance for cultivated wheat, including common wheat. This paper describes the genotyping of nine stable allolines isolated in the offspring from crossing of *T. dicoccum* x *T. aestivum* L. using 21 microsatellite (simple sequence repeats—SSR) markers and two cytoplasmic mitochondrial markers to orf256, rps19-p genes; evaluation of drought tolerance of allolines at different stages of ontogenesis (growth parameters, relative water content, quantum efficiency of Photosystem II, electron transport rate, energy dissipated in Photosystem II); and the study of drought tolerance regulator gene *Dreb-1* with allele-specific PCR (AS-MARKER) and partial sequence analysis. Most allolines differ in genomic composition and *T. dicoccum* introgressions. Four allolines—D-b-05, D-d-05, D-d-05b, and D-41-05—revealed signs of drought tolerance of varying degrees. The more drought tolerant D-41-05 line was also characterized by *Dreb-B1* allele introgression from *T. dicoccum*. A number of non-specific patterns and significant differences in allolines in regulation of physiological parameters in drought conditions is identified. Changes in photosynthetic activity in stress-drought are shown to reflect the level of drought tolerance of the forms studied. The contribution of different combinations of nuclear/cytoplasmic genome and alleles of *Dreb-1 gene* in allolines to the formation of stress tolerance and photosynthetic activity is discussed.

## 1. Introduction

Frequency and intensity of droughts are now increasing as a result of global climate change worldwide. Drought is a major stress that occurs in almost all climatic regions and, as a cause of large grain losses each year, poses a huge challenge to agricultural production in many, especially developing, countries around the world [1,2]. Given the current climate change trend and the potential increase in the world population to 9.6 billion by 2050 [3], global demand for wheat as the main food crop will increase.

The problem of drought tolerance is generally complex and ambiguous. Thus, the term “physiological tolerance” implies survival and viability of the plant, whereas “agronomic tolerance” requires the preservation of economically significant crop under stress conditions [4]. Improving drought tolerance requires certain understanding of mechanisms for genetic control of target traits. However, solving this problem is also difficult since most traits of tolerance in wheat are polygenic and therefore difficult to be interpreted, at both physiological and molecular levels. Therefore, a detailed analysis of both physiological and molecular mechanisms underlying adaptive traits is one of the most important approaches to understanding wheat drought tolerance [5,6].

At the same time, it is already clear that the genetic potential of cultural wheat varieties, which today are used for drought tolerance selection, is apparently insufficient. Despite all the achievements of breeding, world wheat yields have increased by only 1.0% per year over the past two decades [7]. Therefore, the need to involve different species of wheat and its relatives in inter-species crosses is relevant [8]. Indeed, the *Triticeae* tribe is considered to have the enormous potential for stress tolerance [6,9].

The crossing of hexaploid wheat with most species of *Triticeae* resulted in the development of a panel of alloplasmic lines based on replacing nuclear genome of one type with another by re-crossing, when the nuclear genome of wheat was associated with cytoplasmic genomes of species with varying degrees of phylogenetic affinity. Allolines often have many useful morphological, functional, or adaptive features. In addition to being often of interest as a selection material, such lines provide an opportunity to assess the contribution of nuclear and cytoplasmic genomes to regulation of various processes occurring in plants [10].

Given the large amount of data obtained in the study of physiological processes in plants exposed to abiotic stress, it is clear that stress tolerance depends largely on coordination of nuclear genomes and cytoplasm [11,12]. The nucleus as an integrating center of genetic information throughout the cell determines the strategy of species-specific properties formation of organism and the genetic system of cytoplasm determines tactics of vital activity of a cell, controlling the level of adaptive response to any change of environmental conditions [13]. However, rather limited research has focused on the effects of foreign cytoplasm on plant adaptation to drought conditions, and the impacts of cytoplasmic regulatory signals on nuclear genes functioning in plant survival in extreme conditions have not been studied properly.

Photosynthesis, which plays a key role in the formation of plant productivity, is also under complex control of nuclear and chloroplast genes. The study of various nuclear-cytoplasmic chimeras revealed numerous facts of organelle genomes influence on a number of plant functional processes, including photosynthesis-related leaf features and respiratory parameters [14]. On the collection of alloplasmic barley lines, it was shown that the replacement of nucleus on various cytoplasmic backgrounds results in a number of characteristics changes of photosynthetic apparatus—chlorophyll and carotenoid content, number of QB-non-reducing centers of photosystem II, non-photochemical quenching of chlorophyll, etc. [15].

Literature evidence suggests *DREB* and *HSF*s genes are key regulators of wheat complex genetic responses to drought effects and high temperatures. Among the diversity of genes encoding DREB proteins, *Dreb-1* is of the greatest interest [16]. Evolutionarily, many wild species have evolved under extreme environmental conditions (drought, salinity), so they can serve as a source of new genetic variants of *DREB* in breeding wheat with stress resistance.

Previously obtained data on tetraploid forms anatomy and morphophysiology revealed high adaptability of them, in particular the species *T. dicoccum*, to drought and salt stress [17,18]. It is obvious that the study of alloplasmic lines derived from inter-species intersections of *T. aestivum* with drought-tolerant tetraploid species *T. dicoccum* may have both theoretical and practical interest. This work can be considered a methodological case study, whose results will probably help to draw preliminary conclusions about the possible impact of nuclear cytoplasmic interaction on drought tolerance and functioning of photosynthetic apparatus under stress condition. We expect our experiments to reveal the presence of *DREB* gene from *T. dicoccum* in some allolines in the context of genome structure impact on drought tolerance.

## 2. Results

### 2.1. Genotyping of Allolines by Nuclear and Mitochondrial Markers

The scheme for preparation of alloplasmic lines and their morphological description (Appendix A) indicates that the lines may have significant differences in structure of both nuclear and cytoplasmic genomes.

A set of 21 simple sequence repeats (SSR) markers (Appendix A) was used to assess the genetic structure of nuclear genome of allolines. The number of alleles which were detected using a single SSR marker ranged from two to 6six In general, the total evaluation of lines and their comparison with parental genotypes took place across 55 alleles. Among these, five common alleles were only found between the parental forms *T. dicoccum* Schuebl. var. araratum (Host.) and *T. aestivum* Mironowskaya 808 (or M808) variety).

To assess the degree of divergence among alloplasmic wheat lines, a UPGMA dendrogram was constructed based on the results of SSR- analysis (Figure 1). All alloplasmic lines and the parent *T. aestivum* are combined into one large cluster, separately from *T. dicoccum*, the cytoplasm donor. In addition, line D-n-05 is placed into a separate branch from the rest of allolines and M808 with a high level of reliability (butstrep 99). It is also possible to note that there is not only high genetic similarity of D-a-05 and D-b-05 lines (Figure 1) but their morphological characteristics are also similar. (Appendix A).

Thus, analysis of dendrogram of genetic similarity of the studied samples indicates the close relationship of almost all alloplasmic lines (except D-n-05) with M808. In the process of long-term production of alloplasmic lines involving two backcrosses per M808 and 10 generations of self-pollination, the presence of fragments of *T. dicoccum* nuclear genome, with the exception of individual target genes, should be minimized. Nevertheless, line D-f-05 retained genetic material from *T. dicoccum*, labeled Xgwm357 (1AL chromosome), Xgwm192c (4BS), and Xgwm155 (3AL) from *T. dicoccum*. For the same line, unlike the others, there is also a common morphological feature with *T. dicoccum*, namely the manifestation of brown coloring of spike (Appendix A).

Due to the fact that the maternal form in creation of the lines was *T. dicoccum*, the created lines may contain elements of cytoplasmic genome of this species.

The comparative assessment of chloroplast genome was not possible, since the alignment of the known chDNA sequences of hexaploid and tetraploid wheat species in this experiment did not reveal polymorphic SNPs suitable for design of species-specific markers.

To analyze the inheritance of parental mtDNA types, we used polymerase chain reaction (PCR) markers both own and taken from literary sources for two unique genes of mitochondrial genome, *orf256*, *rps19-p* [19,20]. Figure 2 shows the result of restriction endonuclease *Taq* I- digestion of PCR products obtained using the combination of primers orf256f/orf256r. The D-n-05 line is clearly distinguished from other lines by its pattern closest to *T. dicoccum*. For the remaining lines, the pattern of CAPS-marker for *orf256* gene corresponds to variety M808.

Sequencing of the orf256f / orf256r PCR product (~400 bp) of the D-n-05 line showed its complete identity with the analogous mtDNA sequence of *T. timopheevii* (AP013106) and the cytoplasmic male sterility (CMS) line of *T. timopheevii* (♀) x *T. aestivum* (X56186), while there were a number of nucleotide substitutions compared to *T. aestivum* mtDNA (GU985444) (Appendix A).

The result of amplification of the *rps19-p* marker confirms the data on the *orf256* marker.

According to Noyszewski [20], there are two variants of *rps19-p* sequence obtained with rps19f/rps19r primers that differ by deletion of 9 bp (111 bp and 102 bp in length). The 111 bp sequence is characteristic of *T. dicoccum*, while 102 bp sequence is specific for *T. aestivum* (Figure 3). PCR product of D-n-05 line is similar in length to that of *T. dicoccum*, while the rest lines have shortened product characteristic of M808 and *T. aestivum* mtDNA (MH051716). As well as the above- mentioned *orf256* sequence, the 111 bp sequence of *rps19-p* is present in mtDNA of *T. timopheevii* (AP013106) and CMS line of *T. timopheevii* (♀) x *T. aestivum* (X56186) (Appendix A).

Thus, molecular genetic analysis of allolines showed their diversity by content of *T. dicoccum* genetic material in both nuclear and cytoplasmic genomes.

### 2.2. Study of Drought Tolerance and Photosynthetic Activity of Allolines

In order to assess the effect of cytoplasm substitution on drought tolerance, plants were considered in “critical” phases—at the stage of seedlings, when under stressful conditions, there is a vital question of survival of a young plant, and immediately after anthesis, which is important in the formation of grain productivity.

#### 2.2.1. Growth of Seedlings

Experimental comparisons in drought and control conditions based on statistical processing showed that growth parameters per plant varied significantly within allolines having *T. dicoccum* cytoplasm under both optimal and stressful conditions (Table 1).

Under drought stress conditions there were significant differences in allolines D-a-05, D-f-05, and D-n-05 from euplasmic parental forms in primary root growth. The values of the indicator decreased.

But in this case, from the point of view of stress resistance, absolute values of studied indicators were not so much important for us, as were relative (stress/control) values for each alloline. First of all, the relative indicators of root growth because it is the drought tolerance of a root that primarily determines the development of a young plant. The smaller is decrease, the higher is resistance of this alloline. The maximum decrease in growth of the first leaf was noted in D-f-05 line by 63% to the control. D-b-05, D-d-05b, and D-41-05 lines were characterized by the lowest reduction in growth parameters under stress conditions in comparison with controls.

#### 2.2.2. Relative Water Content in the First and Flag Leaves

The results of relative water content (RWC) determination confirmed downward trend in total water content of leaves under stress. Nevertheless, all alloplasmic lines and their euplasmic parental forms studied in this experiment were characterized by relatively small decrease in RWC in the first leaf of seedlings under stress conditions compared to the control (Figure 4).

In the flag-leaf under drought conditions, significant differences from both maternal euplasmic form (*T. aestivum*) and paternal form (*T. dicoccum*) were noted in the downward trends of the feature value at allolines D f-05, Dn-05, Dd-05, D-40-05, and D-42-05. These lines also had maximum reduction in feature values in drought conditions in comparison with control. Allolines D-d-05b (87% to control), Db-05 and D-41-05 (86% to control and at the level of both euplasmic parental forms) also showed a minimal decrease in relative water content of the first leaf of seedlings under drought conditions. After a 72 h period of stress cultivation of cut adult plants, there was a sharp decrease in flag leaf watering in all forms studied. RWC above 50% was only observed in species *T. dicoccum*, *T. aestivum*, and in alloline D-41-05 as well.

#### 2.2.3. Quantum Efficiency of Photochemical Energy Scattering

The data presented in Figure 5 show that plant stress in drought may be accompanied by decrease in the efficiency of light absorbing function of leaves.

Nevertheless, in our experiments, the change in maximum quantum yield of photosystem II (PSII) photochemistry (Fv / Fm ratio) was insignificant in seedlings for all lines, and allolines D-b-05, Dd-05, Dd-05b, and D-41-05 kept this indicator at control level under drought conditions. The change in Fv/Fm ratio of the flag leaf of most lines studied in drought conditions was significant, only species *T. dicoccum* and lines D-d-05b and D-41-05 kept this rate relatively high in drought. In the case of drought conditions were identified allolines that significantly exceed euplasmic form *T. aestivum* according to the indicator.

#### 2.2.4. Electron Transport Rate Through PSII

The first leaves of *T. dicoccum*, D-b-05, D-d-05b, and D-41-05 seedlings maintained relatively high speed of non-cyclic electron transport in conditions of induced drought. Non-cyclic electron transport rate through PSII (electron transport rate—ETR) values in drought were at control level (Figure 6a). However, only in the flag leaves of plants *T. dicoccum* ETR in drought was 18% higher than control values. Under stress conditions the level of this indicator in allolines decreased significantly. The highest ETR values among allolines were observed in D-41-05-line flag-leaf (69.6% to control).

#### 2.2.5. Changing of the Quantum Yield of Energy Dissipated in PSII

Quantum yield of regulated energy dissipated in PSII Y(NPQ) under stress conditions tended to decrease (Figure 6b). However, allolines seedling D-d-05b, D-n-05, and D-41-05 Y(NPQ) values remained at control levels in drought. In the flag leaves of intact plants allolines D-d-05 and D-41-05, the values of this indicator in drought significantly increased relative to control.

The parameter of the quantum yield of non-regulated energy dissipated in PSII Y(NO) under conditions of stress in drought increased in euplasmic parental forms and in practically all studied allolines as well. This occurred both at the stage of the first leaf and at measurements on flag leaves (Figure 6c). D-41-05 line showed different trend where the value of this indicator remained at the level of control at both the early and late stages of ontogenesis.

Thus, on the basis of physiological indices for four allolines D-b-05, D-d-05, D-d-05b, and D-41-05 revealed signs of drought tolerance of varying degrees. Among them, alloplasmic line D-41-05 was the most stable.

### 2.3. Identification of Gene Regulating Drought Tolerance in Alloplasmic Lines

We searched for potential genes that could be responsible for the presence of drought tolerance characteristics in alloplasmic lines.

The *Dreb-1* gene is a representative of a big family of transcription factors, regulating drought tolerance. AS-PCR marker earlier developed to the gene [21] allows us to efficiently discriminate two alleles of the *Dreb-B1* gene. One of these alleles characterizes tetraploid wheats that have BA- genome, like *T. dicoccum*, and differs from the second allele which is specific for *T. aestivum* by only one SNP in the first exon. AS-PCR marker was used in analysis of alloplasmic lines and their parents. As a result, we showed the presence of band characteristic of *T. dicoccum* in two lines, one of which (D-41-05) displays the highest level of drought tolerance (see above) (Figure 7).

This fact implies that, in these lines, there was an introgression of *Dreb-B1* gene from nuclear genome of *T. dicoccum* into genome of *T. aestivum* (M808). To confirm this introgression, we sequenced a part of the PCR product containing the characteristic SNP (see Materials and Methods) and found the same substitution C→A in the DNA sequences isolated from *T. dicoccum* and two above-mentioned lines (Appendix A)

## 3. Discussion

It was shown that differences between alloplasmic lines across a number of morphophysiological and photosynthetic parameters affect such an important trait as stress tolerance, particularly drought tolerance.

The general effect of abiotic stresses on plants can be summed up as a disturbance of water and growth inhibition by decrease stretching of dividing cells. [22]. In cereals, yield is determined by root first. The main strategy of plant adaptation to drought is implemented through a number of physiological mechanisms that maintain a gradient of water potential between soil solution and root cells. This allows a root to continue absorption of soil water under drought conditions [23]. Allolines of Db-05, Dd-05b, and D-41-05, which in our experiments had the least reduction of root system growth and leaf length in drought with respect to controls, can be considered the most tolerant to drought in the early stages of ontogenesis, whereas allolines of Df-05 and Dn-05 showed the least tolerance.

In winter wheat, the tolerance of seedlings to drought means first of all their survival and ability to grow further, while the ability of plants to carry drought at the flag-leaf stage can affect grain productivity. If we study both seedlings and flag-leaf reaction to drought, we can obtain more complete characterization of drought tolerance of observed allolines.

Reduction of leaf water content and growth cessation in the field are important indicators of drought sensitivity; RWC reflects the balance between supply of water to leaf tissue and rate of transpiration [24]. The decrease in RWC due to drought may depend on energy decrease and it has been observed in many plants [25,26]. The tendency to water content reduction in leaves under stress was confirmed by the results of determining the relative water content on both the first seedlings sheet and the flag leaves stages. However, the results of the experiment show that adult plants placed in the examination solution experienced more stress than seedlings, and the loss of water by the leaves was much higher. Nevertheless, alloline D-41-05 in this experiment, as it did in the previous ones, showed greater drought tolerance than the rest on both early and late stages of ontogenesis.

Efeoglu et al. [27] indicated that drought is an important factor responsible for slowing down not only plant growth but also for photosynthesis reduction. Photosynthetic responses to drought are complex and depend on both intensity and duration of stress and the stage of plant development. In seedlings experiencing osmotic stress, it is possible to suppress chlorophyll biosynthesis and reduce the synthesis and assembly of PSI and PSII light collection complexes, as well as adapt plants to avoid excess light absorption, which is harmful. On the late stages of ontogenesis, functional photosynthetic complexes are already formed in the leaves, and water stress causes the formation of AFCs due to excessive light absorption, which affects the photosynthetic apparatus [28]. We assume that the intensity of flag leaf photosynthesis in sudden and severe drought conditions limits the parameters of water regime.

The Fv/Fm indicator represents the maximum potential quantum efficiency of PSII if all capable reaction centers have been discovered [29]. In general, the more stress plants experience, the fewer open reaction centers they have. The values of this parameter in response to the impact of drought in seedlings have not changed significantly in our experiment. This fact testifies the preservation of operability of reaction centers of PSII of all the allolines studied. Substantial change in the Fv/Fm ratio in flag leaves of most lines studied in drought is an indicator of photo inhibition [30]. The *T. dicoccum* and lines D-d-05b and D-41-05, which maintain relatively high level of this indicator, show the greatest drought tolerance.

Although the decrease in ETR values in most allolines at different stages of ontogenesis may be due to activation of non-photochemical extinguishing mechanisms, the maximum electron transport rates may be an indicator of photosynthetic activity [31]. Thus, *T. dicoccum* species and D-41-05 line from the experiment are the most drought tolerant of all the samples studied for this indicator. At the same time, both control and stress conditions revealed clear reliable differences in most studied allolines in the upward direction of ETR value referred to euplasmic form *T. aestivum*, and in the downward direction of ETR value referred to euplasmic form *T. dicoccum*.

By quenching analysis, one of the key parameters—quantum output of light-induced non-photochemical fluorescence extinguishing Y(NPQ)—has been identified, which is the fastest reversible heat scattering process of absorbed light energy in PSII antenna [32,33]. The maximum value Y(NPQ) is observed in drought-tolerant alloline D-41-05.

If any increase in Y(NPQ) expresses an attempt to dissipate excess energy, increase in quantum output of non-regulated heat dissipation and fluorescence Y(NO) radiation means that flows of excess energy are out of control and can cause photopotection of plants [34]. As has been shown in *Arabidopsis* plants, any increase in non-photochemical component can be associated with inclusion of protective mechanisms in short-term stress [34]. High Y(NO) values in plants can be a sign of serious problems with the redistribution of excess light energy entering the PSII. In our experiment, both on seedlings and on flag leaves, the value of this indicator almost in all lines except D-41-05 increased referring to euplasmic form of *T. dicoccum*. These forms might have disorders in the work of PSII or even in its structure, and D-41-05 line may be considered the most drought-tolerant in both early and late stages of ontogenesis.

In general, the commonality of main non-specific mechanisms of response of studied forms of wheat to the effects of drought is shown. There is a significant species-specific and sort-specific diversity of individual reactions of each line due to the genotype.

The dendrogram of genetic similarity of studied alloplasmic lines (Figure 4) indicates close relationship of eight out of nine lines with Mironovskaya 808. Lines D-N-05 significantly differ both from M808 and *T. dicoccum.* This fact implies relatively higher degree of genetic exchanges between the parental genomes in these lines in comparison with the rest. Even within a group of lines close to the paternal M808 genotype, most lines, such as D-41-05, D-40-05, D-f-05, D-d-05, and D-42-05, are quite isolated from each other. The data obtained indicate different mechanisms of variability and subsequent stabilization of nuclear genome during formation of alloplasmic lines. This leads to a wide range of observed phenotypic and biochemical characteristics among the studied lines.

Nuclear cytoplasmic interactions, in particular, the influence of the mitochondrial genome, also make a great impact to the phenotypic variability, fertility, and viability of hybrids [35]. In this work, analysis of mtDNA of alloplasmic wheat lines using molecular markers showed that eight out of nine lines have the patterns of the used PCR and cleaved amplified polymorphic sequence (CAPS) markers similar to those of *T. aestivum* (M808) (Figure 5 and Figure 6). Consequently, the restoration of fertility of alloplasmic lines under study correlates with the male parent type of mtDNA which substitutes maternal type during backcrossing. This result agrees well with the previous data obtained using *Hordeum* x *Triticum* alloplasmic hybrids [36]. An exception is the D-n-05 line, in which the patterns of markers for the *rps19-p* and *orf256* genes were similar to the parental species *T. dicoccum*. Interestingly, *orf256* sequence in this line was identical with the analogous sequence of CMS line *T. timopheevii* (♀) x *T. aestivum* (X56186). However, unlike the latter, the D-n-05 line is fertile. Hence, the D-n-05 line is a perspective model to search fertility restoration genes in the nuclear genome. In terms of drought tolerance, the latter line showed the greatest sensitivity to drought, which can be explained by nuclear-cytoplasmic imbalance and, as a result, reduced viability under stress.

It is a well-known fact that chloroplast DNA (chDNA) contains several genes that control the process of photosynthesis. Therefore, it was logical to assume that observed differences between alloplasmic lines in terms of the rate of photochemical reactions are due to specific interactions of nuclear and chloroplast genomes. Previously, complete chloroplast DNA sequences of different cereal species, both diploid and polyploid, were obtained [37], and a few sequences were effectively used as barcodes of cytoplasm of certain predominantly alien species [38]. However, in our case, it seems not possible to assess the impact of various parental types of chDNA within lines under study, since an alignment of the known chDNA sequences of hexaploid and tetraploid wheat species did not reveal polymorphic SNPs suitable for the design of species-specific markers.

In this study, we paid special attention to *DREB* gene. *DREB* gene encodes the transcription factor *DREB* defining answer of plant to some abiotic stress and it is one of the most studied drought tolerance markers. Thus, in experiments by Pellegrineschi et al. [39], stress-induced expression of *DREB1A* from Arabidopsis increased drought tolerance of wheat, indicating the promise of using *DREB* in improving wheat adaptation to drought action. Transcription factors associated with dehydration of sensitive elements (*DREB*) have been reported to increase drought tolerance of transgenic wheat [39,40]. Higher drought tolerance of transgenic plants *AtDREB1A* has been achieved by increasing their relative water, chlorophyll, sugar and proline content compared to non-transgenic plants [41]. Members of two different families of transcriptional factors, *TaDREB5* and *TaNFYC-A7*, were found to be associated with drought yields [42,43,44]. Various studies have demonstrated that improved stress tolerance by over-expression of *Dreb-1* genes is associated with sustained photochemical efficiency and photosynthetic capacity as compared with wild-type plants [45,46,47].

So, we used a previously developed marker [48] to analyze whether this gene relates to altered physiological traits due to inheritance of various *Dreb-1* alleles from parental species. The results demonstrated the introgression of allele *Dreb-B1* from maternal species *T. dicoccum* into the paternal nuclear genome of two lines, D-41-05 and D-f-05 (Figure 7). The former line has the highest drought tolerance among all the studied lines, while the latter has an intermediate level. Since these lines have a different location on the dendrogram (Figure 4) implying various rate of interspecific recombination, the difference in drought tolerance can be explained by the effects of other genes, particularly, those differentially inherited from maternal genome. It should be noted that the mode of functioning of *Dreb-1* has not yet been studied. In the future, we plan to analyze the expression of *Dreb-1* in alloplasmic lines under the influence of drought.

Expression and regulation mechanisms controlling a wide range of photosynthesis genes in moisture-deficient environments require certainly more attention, since understanding their stress response mechanisms can significantly improve the efficiency of creating varieties with increased adaptability to arid conditions. It is integrated approach that can provide the most favorable potential basis for future wheat breeding programs.

## 4. Materials and Methods

Nine alloplasmic wheat lines derived from the crossing of *T. dicoccum* var Araratum (Host.) and *T. aestivum*, Mironowskaya 808 (M808) variety, followed by multi-year selection (F12) held by Professor N.A. Khaylenko, were chosen as material for the study.

### 4.1. DNA Isolation and Analysis of SSR-Markers

Total genomic DNA was extracted from seven-day-old seedlings [49]. DNA concentration was determined using a SmartSpec TM Plus spectrophotometer (BioRad, Hercules, California, USA).

In the work 21 SSR markers were used selected so as to cover maximally the wheat genome, including Xgwm334, Xgwm437, Xgwm18, Xgwm357, Xgwm613, Xgwm3, Xtaglgap, Xgwm408, Xgwm577, Xgwm261, Xgwm95, Xgwm155, Xgwm186, Xgwm130, Xgwm389, Xgwm513, Xgwm160, Xgwm192, Xgwm148, Xgpw2255, Xgwm190, and Xgwm469. The PCR procedure was published in Roeder et al. [50]. PCR fragments were separated on ABI PRISM 3100 automatic sequencer (Applied Biosystems, Waltham, MA, USA). Fragment size was calculated using the ABI GeneScan software, version 2.1 (Applied Biosystems, Waltham, MA, USA).

SSR analysis data were used to study the genetic similarity of lines using PHYLIP software package (Version 3.69, Seattle, Washington, USA) [51]. To assess the reliability of constructed trees, bootstrap analysis was performed for 100 replicates.

### 4.2. Mitochondrial Genome Analysis

The following genes were used for the analysis of mtDNA of alloplasmic lines: 1) *orf*256- chimeric reading frame near the cytochrome oxidase gene (associated with cytoplasmic male sterility (CMS) [19]; 2) *rps*19-p- pseudogene encoding a ribosomal protein [20]. Corresponding primer pairs for each gene were the following: orf256f (ggaagggaaccaatcaagtcacc)/orf256r (gatcctgctcgtaaaggctcag); rps19f (tgctccgtactcatttacaatgg)/rps19r (atagggtcttcgtctccgtg).

PCR mixture contains 2 mM MgCl_2_ and other components in standard concentration. The following PCR program was used: initial denaturation at 94 °C for 2 min; 35 cycles of 94 °C for 45 sec, 55–60 °C for 45 sec, 72 °C for 10–40 sec; and a final extension at 72 °C for 2 min. PCR products were electrophoresed on 2% agarose gels, stained with ethidium bromide and visualized under UV light. *orf*256 PCR product was excised from gel, purified using a QIAquick PCR purification kit (QIAGEN, Hilden, Germany) and sequenced using a bigdye terminator v3.1 cycle sequencing kit (Applied Biosystems, Waltham, MA, USA). Sequencing products were analyzed at the Collective Use Center “Genomika” of the SB RAS. Additionally, in the case of *orf*256, cleaved amplified polymorphic sequence (CAPS) marker was used as follows: the PCR product was digested with *Taq* I restriction endonuclease (Sibenzyme, Novosibirsk, Russia), followed by electrophoresis on 2% agarose gel.

### 4.3. Analysis of the Drought Tolerance Regulator Gene Dreb-1

In this work, *Dreb-1* allele-specific PCR (AS-PCR) marker described in Wei et al. [48] was used. AS-PCR amplification was performed in a total volume of 25 µL containing 100 ng of genomic DNA, PCR reaction buffer, 0.25 µM of each primer (consensus primer P18R: ttgtgctcctcatgggtactt and an allele-specific primer P40: atatggattgccttgatgca), 0.45 mM of each deoxyribonucleotide, 4.0 mM MgCl_2_ and 1.6 U of *Taq* DNA polymerase. PCR was carried out with the following program: initial denaturation at 94 °C for 3 min; 35 cycles of 94 °C for 1 min, 56 °C for 1 min, 72 °C for 1.5 min; and a final extension at 72 °C for 10 min. The PCR products were detected using 2.5% agarose gel. To confirm the SNP characteristic of *T. dicoccum Dreb-B1* gene, PCR products obtained with primers P18F (cccaacccaagtgataataatct) and P18R (see above) were digested with restriction endonuclease *Bst*F5I (Sibenzyme) followed by agarose gel electrophoresis. 635 bp fragment specific for *Dreb-B1* was excised from the gel and partially sequenced using P18F primer.

### 4.4. Analysis of Physiological Parameters of Drought Tolerance

During the seedling stage, all alloplasmic lines and corresponding euplasmic controls were subjected to simulated drought: at first seeds of studied wheat species and lines were germinated in thermostat, in the dark at 25 °C for three days. Thereafter, all seedlings were grown for five days in water crops at 26 ± 2 °C at light 657 μmol and 16-hour photoperiod. Then for 72 h at 26 ± 2 °C and light 657 μmol and 16-hour photoperiod some of them (25 plants in each biological replicas) were exposed to 17.6% sucrose solution (*w*/*v*). These sucrose concentrations caused visual differences in biomass growth and accumulation samples when growth of drought sensitive seedlings reached 40–50% of control values. The control was on the rest of the plants growing under similar temperature and lighting conditions as experimental ones, but in water [52].

Prior to the flag-leaf stage, plants were grown under field conditions. To conduct the experiment on the effects of osmotic stress on a flag-leaf several plants (three plants in each biological replica for control and experience) were cut off immediately after pollination, transferred to laboratory to stress-solution conditions for 72 h described above on seedlings. Plants exposed under similar temperature and lighting conditions on water also served as controls.

The RWC of the leaf plates was calculated by the following formula:

RWC = ((a − b)/a) × 100%, where a–the initial mass of leaves, mg; b–weight of leaves after drying at 105 °C, mg.

All experiments were held at least three time replicates.

### 4.5. Analysis of Photosynthetic Parameters

The determination of quantum yield of fluorescence, fixed rate of electron transport through photosystem II (ETR) was carried out in recording mode of light curve on fluorometer Junior-PAM (Chlorophyll Fluorometer, “Heinz WalzGmbH”, Germany) at wavelength 450 nm. The value of utilization of absorbed photons antennas PS II during electron transport and thermal dissipation were determined using the quantum efficiency (Y) of photochemical energy dissipation (Y(II)), light dependent (Y(NPQ)) and light independent (Y(NO)) thermal scattering. The state of photosynthetic apparatus was determined on the basis of these parameters, according to Baker [53]. Values of photosynthetic parameters were calculated using software ImagingWinv2.41a (Walz). Charts based on the results were compiled. Processing of data obtained on fluorometer, and graphing was performed using Microsoft Excel. Obtained data were saved in spreadsheet format. Atypical values were excluded from the data based on t-tests, the standard error of the average sample was calculated. To assess photosynthetic activity (PA) of a leaf there was taken into account the region of its middle third since it has the most homogeneous PA intensity. All experiments were held at least three time replicas.

## 5. Conclusions

The results obtained in this methodological study show similar non-specific allolines responses under stress conditions in both absolute and relative values and suggest that parameters of photosynthetic leaf activity are fully correspondent to the tolerance of these drought stress lines.

The conducted assessment of morphophysiological and photosynthetic parameters (stress/control) suggests alloline of D-41-05 as the most drought-tolerant form studied.

Introgression of *Dreb-B1* allele from maternal species *T. dicoccum* for two lines (D-41-05 and D-f-05) was demonstrated and confirmed, one of which (D-41-05) showed high level of drought tolerance.

It is noted that combination in the alloplasmic line of nucleus and cytoplasm, originating from parental forms belonging to different species, can both improve and degrade important physiological parameters of stress tolerance and photosynthetic activity, which requires further molecular genetic analysis.

## Figures and Tables

**Figure 1 ijms-21-03356-f001:**
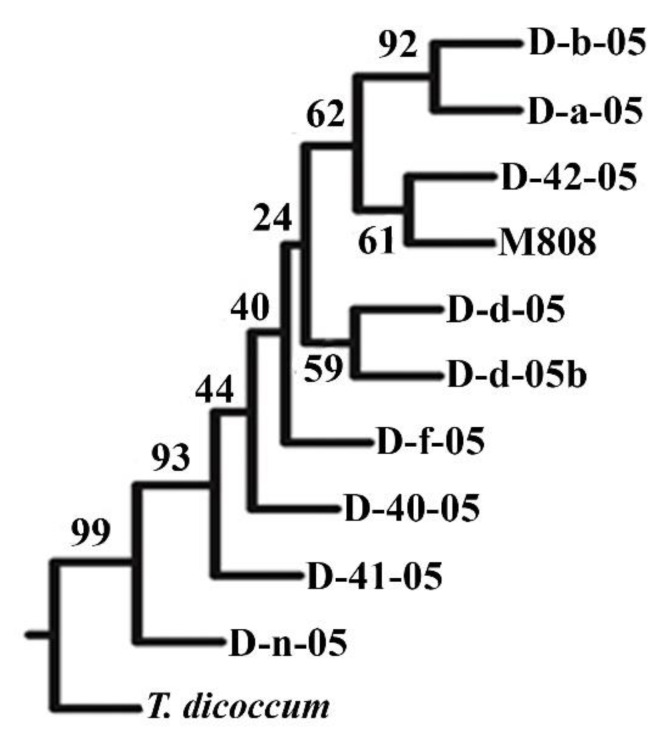
UPGMA-dendrogram of genetic similarity of alloplasmic wheat lines according to simple sequence repeats (SSR) analysis. Designations of parental forms: *T. aestivum* cv. Mironovskaya 808 (M808), *T. dicoccum*.

**Figure 2 ijms-21-03356-f002:**
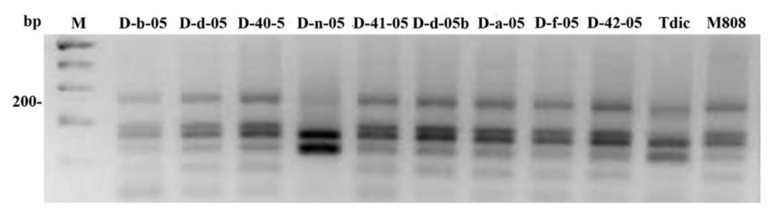
Genotyping of alloplasmic lines with *orf256* CAPS-marker. The polymerase chain reaction (PCR) products obtained using specific primers orf256f/orf256r were digested by *Taq* I- endonuclease. The “100 bp” ladder was used as a length marker.

**Figure 3 ijms-21-03356-f003:**
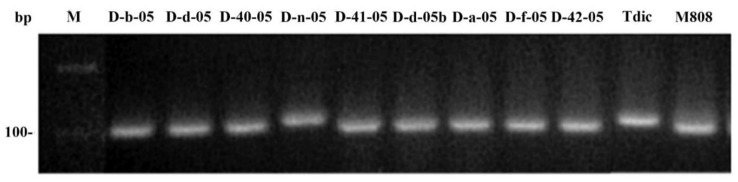
PCR of genomic DNA of alloplasmic lines with primers rps19f/rps19r.

**Figure 4 ijms-21-03356-f004:**
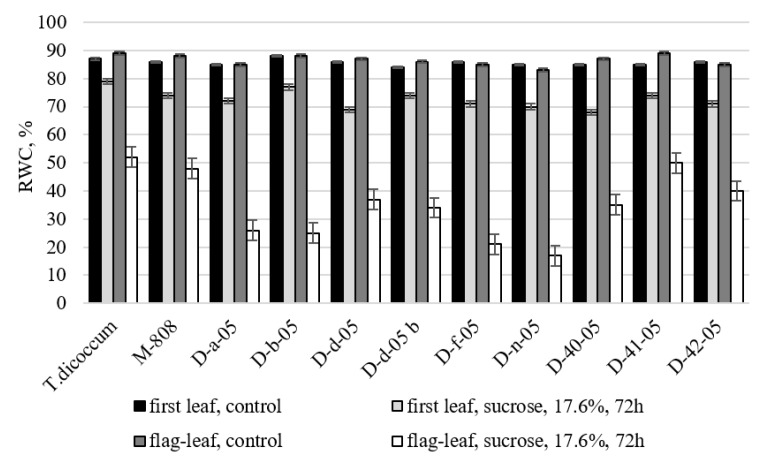
Change in relative water content of alloplasmic lines under drought conditions (17.6% sucrose, 72 h). Values presented are means (± SD).

**Figure 5 ijms-21-03356-f005:**
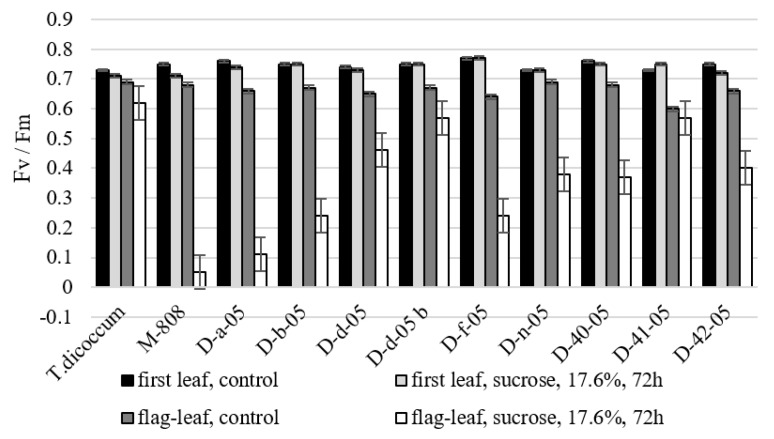
Changes of the maximum photochemical efficiency of PSII (Fv/Fm) of alloplasmic lines under drought conditions (17.6% sucrose, 72 h). Values presented are means (± SD).

**Figure 6 ijms-21-03356-f006:**
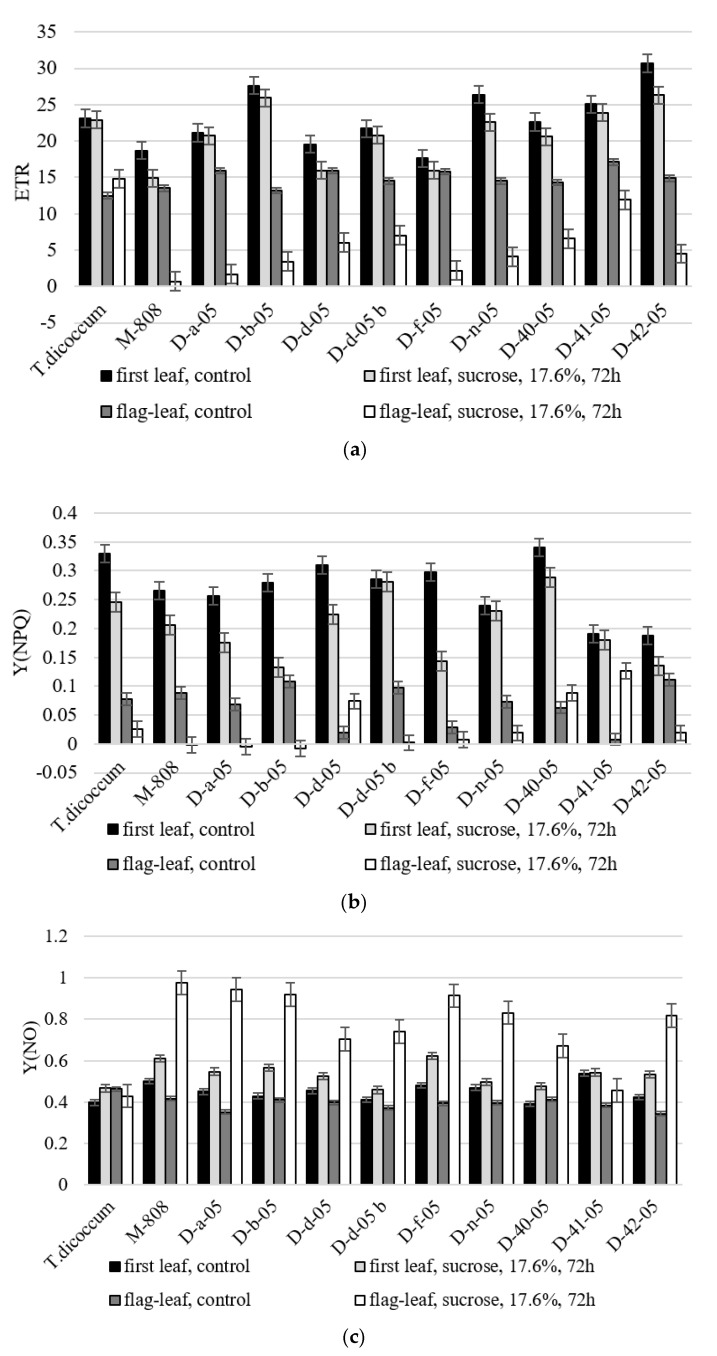
Mean of photosynthetic parameters electron transport rate (ETR) (**a**), Y(NPQ) (**b**), Y(NO) (**c**) (relative units) of alloplasmic lines under optimal and drought conditions (17.6% sucrose, 72 h). Values presented are means (± SD).

**Figure 7 ijms-21-03356-f007:**
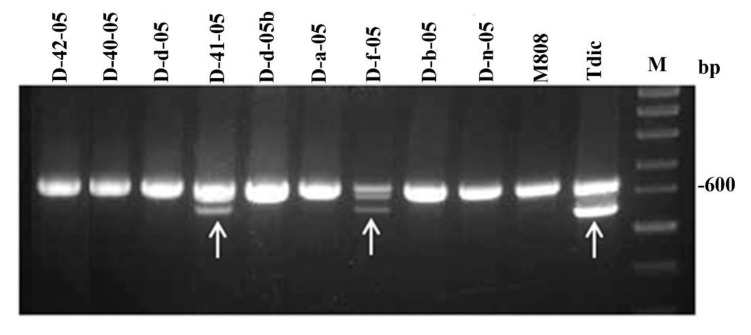
PCR with a combination of primers P40/P18R for identification of the *Dreb-1* gene. The arrows indicate the common band for lines D-41-05, D-f-05, and *T. dicoccum*, corresponding to the *T. dicoccum-* specific allele of *Dreb-B1*. The upper double band corresponds to the *Dreb-A1* and *Dreb-D1* genes.

**Table 1 ijms-21-03356-t001:** Growth parameters of the first leaf and primary roots of seedlings of alloplasmic wheat lines and their parental forms under drought conditions (17.6% sucrose, 72 h) (means ± standard deviations).

Species, Line	Root, cm			Leaf, cm		
	Control	Stress	% to control	Control	Stress	% to control
*T. aestivum*	8.22 ± 1.53	6.86 ± 0.81	83	15.42 ± 1.15	13.3 ± 1.04	86
*T. dicoccum*	7.03 ± 0.65	6.58 ± 0.85	94	16.92 ± 1.10	13.99 ± 1.21	83
D-a-05	7.46 ± 0.61	4.32 ± 0.76 ^ad^*	58	14.74 ± 1.11	11.83 ± 1.06	80
D-b-05	7.81 ± 1.66	7.04 ± 0.96	90	14.66 ± 1.42	14.02 ± 1.28	95
D-f-05	7.43 ± 0.84	4.71 ± 0.62 ^a^*	63	17.60 ± 1.19	11.10 ± 1.11 *	63
D-n-05	7.65 ± 1.13	4.30 ± 0.91 ^ad^*	56	14.26 ± 1.30	11.04 ± 1.33	77
D-d-05	10.64 ± 0.84	6.76 ± 0.78 *	64	15.61 ± 1.50	12.74 ± 1.71	82
D-d-05 b	9.39 ± 1.99	8.17 ± 0.71	87	17.20 ± 1.61	14.43 ± 1.71	83
D-40-05	9.67 ± 1.37	7.45 ± 1.12	77	16.46 ± 1.37	14.24 ± 1.32	87
D-41-05	7.26 ± 1.07	6.43 ± 1.27	88	15.69 ± 1.30	13.10 ± 1.11	83
D-42-05	9.26 ± 1.33	6.33 ± 0.92	68	15.91 ± 0.97	12.63 ± 1.37	79

Note: * indicate significant differences to control, ^a^ indicate significant differences to *T. aestivum*, and ^d^ indicate significant differences to *T. dicoccum* at *p* ≤ 0.05.

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
