# Peer review of "Drought Stress Tolerance and Photosynthetic Activity of Alloplasmic Lines T. dicoccum x T. aestivum"

_ijms, 2020, doi:10.3390/ijms21093356_

Round 1

Reviewer 1 Report

The manuscript presents an experiment relating to an increase of genetic diversity of common wheat by crossing it with T. dicoccum. The work is interesting, but it requires many corrections (major revision).

Introduction

 The introduction should include information about the role of proteins encoded by the Dreb genes, which is why the authors paid attention to these proteins. The family name Triticeae should be in italics.

Page 2 line 60-61 repetition of earlier information;

line 62 delete ‘In this case’ .. Line 64 delete ‘At the same time’; line 72 - 76 bad style

Results

Chlorophyll fluorescence parameters are not the best to show greater drought tolerance. The D-41-05 genotype chosen by the authors does not show higher values ​​of Fv / Fm, ETR parameters than T. dicoccum, and all the studied genotypes are better in this respect than T. aestivum. RWC is the same in the parental forms. Also, the length of the roots and leaves does not show that this genotype really stands out from the others. Table 1 lacks the units in which the leaves and roots were measured.

Discussion

I would suggest the authors look again at the results as a method that can be used. In my opinion, on the base of presented data, the selected genotype is unlikely to meet the criterion of greater drought tolerance. The pressure on growth parameters such as leaf surface or root length should be increased, and why were the fresh and dry mass of the roots and aboveground part of plants not given? Maybe the authors have these data and they could provide them as a supplement to RWC. In my experience, dry plant weight is the best parameter for drought tolerance, and among parameters of chlorophyll fluorescence PI - performance index, which provides information about the actual PS2 photosynthetic, is the most useful in the selection of plants with higher drought tolerance (efficiency (Fv / Fm is the potential photochemical performance of PS2 and it often does not reflect the degree of plant stress response).

Methods

Section 4.4 must be improved. There is a lot of ambiguity. When plants were cultivated under drought stress? All the time? Seed germination conditions, and under what conditions did seedlings grow 5 days after germination? Were the plants grown in hydroponics until heading? Why were plants cut after pollination? It is unclear what happened with these plants and why they were taken to the laboratory?

Conclusion (error in chapter name)

You don't start with “Thus”

lines 447-449 sentence must be corrected, it is unclear

References

The list of references must be tailored to the requirements of the journal

Author Response

Thank you very much for remarks.
We have made appropriate necessary additions to all sections now.
All corrections and additions in the text are marked in yellow.

But indeed, we need clarity in some detals.

From the point of view of physiology of stress resistance absolute values in the listed indicators were not so much important for us, as relative values were, i.e. how significantly the value of the listed indicators in stress conditions in a alloline decreases relatively to optimal (control) conditions for it. The smaller is the decrease, the higher the resistance of this alloline. This applies primarily to growth parameters. At this stage of the experiment, several lines of D-b-05, D-b-05b and D-41-05 were identified by the relative indicators of growth under stressful conditions to control ones and first of all - by relative indicators of root growth, because it is the drought tolerance of the root that primarily determines the development of the young plant. Of course, we compare values of signs in the lines relative to euplasmic parental forms. Thus, the root system gain (% stress/control) of the D-41-05 line was slightly lower than that of T. dicoccum, but higher than that of Ðœ808. As for ETR - at the seedling stage the values of D-41-05 at stress is at the level of control and close to the values of T. dicoccum, at the flag-leaf stage we see a decrease in the values of the sign both relative to control and relative to T. dicoccum, but this decrease is less pronounced than in other lines, and the value of the sign at stress is much higher than in Ðœ808. Fv/Fm values of D-41-05, at the stage of seedlings under stress conditions exceed those as own control and stress in T. dicoccum. At the stage of flag-leaf are also related to those in drought-tolerant species T. dicoccum and much higher than in Ðœ808. Exceeding the sign value over the Ðœ808 is more pronounced than the rest of the allolines except for the D-d-05b line, which we also noted as relatively drought tolerant. Thus, the conducted assessment of morphophysiological and photosynthetic parameters suggests alloline of D-41-05 be considered the most drought-tolerant form studied by a complex of signs and first of all - in comparison with stress values relatively to its control values as we say above.

Yes, we have biomass data. Raw biomass was large in control (the highest) but dry biomass in control was low, whereas dry biomass in the experiment was high, and fresh biomass was low. As with RWC, the difference on seedlings between genotypes was insignificant, but the difference on the flag sheet was very tangible. But, in general, the picture on RWC we see in the context of this article is more informative than the picture on biomass. Water retention in drought is an important physiological sign. Flag leaves with low RWC at stress were virtually dry and their photosynthetic activity was accordingly very low. Lines with higher RWC also retained the ability to photosynthesis. But in future we will certainly pay closer attention to biomass analysis. Thank you very much for this advice.

As the literature shows, the fluorescence of chlorophyll is considered to be an important indicator of stress tolerance in different species and variants of plants, for example, durum, wheel and tobacco. This is confirmed by the results of our previous studies. With regard to the legality of the use of chlorophyll fluorescence parameters in this work as criteria for drought resistance of the lines studied, we can answer that this work is experimental and is aimed, inter alia, at confirming or refuting this possibility.

We thank the distinguished reviewer for this observation and certainly we will also take it into account in future studies. Yes, indeed, Fv/Fm indicator does not always reflect the degree of response of plants to stress, as there is a number of evidence in the literature, but Fv/Fm is also considered a fast-measuring indicator of stress in plants. Our work supports the utility of this parameter, in addition to other fluorescence measurements in stress tolerance screenings. We took measurements of a number of photosynthetic parameters as far as the technical potential of JUNIOR-PAM allowed, and in this work placed emphasis (besides Fv/Fm indicator) on ETR parameter which in our opinion was very informative and also on Y(NPQ) and Y(NO) which, in a number of literary data and our opinion, are important exponents of stability/sensitivity of photosynthetic device in stressful conditions.

The concentration of stress-agent (sucrose) in our work is based on recommendations of the VIR named after Vavilov and has been worked out on wheat many times in various experiments. It is based on reduction of growth parameters in forms unstable to osmotic stress by more than 50%.

Reviewer 2 Report

This work is about the utilization of the genetic background of T. dicoccum for the increase of drought tolerance in modern T. aestivum varieties. Several parameters were investigated, but the work itself seems to be rather a methodological case study, than a step towards selecting a successful alloline, as the evaluation of them might be insufficient for this aim.

General comments

I missed the detailed evaluation of the allolines from the ms, also, the results of the investigated parameters are not discussed in details. However, if the aim was the demonstration of the presence of DREB genes, and the effect of them on the investigated parameters, then it is not necessary to elaborate on this part. At the same time, the emphasis should be put on this fact, i.e. this research is a methodological study.

With regards to growth parameters, i see, that all of the allolines performed worse, or slightly better, than the parental ones. I suggest to discuss this part a bit more detailed.

Detailed comments

line 14: potential source

keywords: T. diccocum is also used in the title of the MS, choose an other keyword

line 43: solving this problem

line 73: it was shown, that

line 97: wheat lines, a UPGMA dendrogram was constructed

line 155: Growth mistyped

Table 1: I see, that almost all allolines performed worse, than the parental forms, although this difference is not significant in most cases. I think, more emphasis should be put on this here.

line 254: drought sensitivity

line 349: used a previously developed

line 387: The following PCR program was used:

line 441: Conclusion

line 446: alloline of D-41-05 to be considered as the most

Author Response

Thank you very much for remarks.

We have made appropriate additions to "Introduction", "Results" and "Conclution" sections now.

All  errors indicated by you are corrected.

All corrections and additions in the text are marked in yellow.

Round 2

Reviewer 1 Report

The authors have corrected the text a bit, however, there are minor mistakes.

Line 422 tangled up the letter of the Russian alphabet "i"

line 428 "plant material" replace with "plants"

line 426 "on water" replace with "in water"

line 436  Replace replicas with replicates

lux is an illegal unit, exchanged for μE or μmol (photons)

Table 1 what is the unit "sm" probably “cm”

It is still unknown why the plants in the field were cut before pollination and what analyzes were carried out on them

Author Response

Thank you for remarks.

All mistakes indicated by you are corrected and marked in green now.

In winter wheat the tolerance of seedlings to drought means first of all their survival and ability to grow further, while the ability of plants to carry drought at flag-leaf stage can affect grain productivity.  The flag-leaf responds to an emerging water shortage, especially at the developmental stage immediately after pollination, and it is directly related to the wheat production process. If we will study both seedlings and flag-leaf reaction to drought, we can obtain more complete characterization of drought tolerance of observed allolines (lines 260-263).  Therefore, the first stage of our physiological experiments was the study of drought tolerance of seedlings. The second stage is the study of the parameters of RVC, Fv / Fm, ETR, Y(NO) and Y(NPQ) on the flag-leaf of allolinies at the stage immediately after pollination. Thus, the same experiments were carried out on the flag-leaf as on the seedlings with the exception of measuring growth parameters. All data are presented in graphs. Experimental plants were cut and transferred for laboratory tests in order to quickly create severe stress directed at the flag sheet and get visual results. In the field, this is not possible.